# Parallel Broadband Femtosecond Reflection Spectroscopy at a Soft X-Ray Free-Electron Laser

**Robin Y. Engel** [1,2] , **Piter S. Miedema** [1] , **Diego Turenne** [3], **Igor Vaskivskyi** [3,4], **Günter Brenner** [1], **Siarhei Dziarzhytski** [1] , **Marion Kuhlmann** [1], **Jan O. Schunck** [1,2] , **Florian Döring** [5], **Andriy Styervoyedov** [6], **Stuart S.P. Parkin** [6], **Christian David** [5], **Christian Schüßler-Langeheine** [7], **Hermann A. Dürr** [3] and **Martin Beye** [1,2,*]

1   Deutsches Elektronen Synchrotron, 22607 Hamburg, Germany; robin.engel@desy.de (R.Y.E.); piter.miedema@desy.de (P.S.M.); guenter.brenner@desy.de (G.B.); siarhei.dziarzhytski@desy.de (S.D.); marion.kuhlmann@desy.de (M.K.); jan.schunck@desy.de (J.O.S.)
2   Department of Physics, University of Hamburg, 20355 Hamburg, Germany
3   Department of Physics and Astronomy, Uppsala University, S-75120 Uppsala, Sweden; diego.turenne@physics.uu.se (D.T.); igor.vaskivskyi@physics.uu.se (I.V.); hermann.durr@physics.uu.se (H.A.D.)
4   Center for Memory and Recording Research, University of California San Diego, 9500 Gilman Drive, La Jolla, CA 92093-0401, USA
5   Paul Scherrer Institut, 5232 Villigen-PSI, Switzerland; florian.doering@psi.ch (F.D.); christian.david@psi.ch (C.D.)
6   Max-Planck Institut für Mikrostrukturphysik, Weinberg 2, 06108-06132 Halle, Germany; andriy.styervoyedov@mpi-halle.mpg.de (A.S.); stuart.parkin@mpi-halle.mpg.de (S.S.P.P.)
7   Helmholtz-Zentrum Berlin für Materialien und Energie GmbH, 12489 Berlin, Germany; christian.schuessler@helmholtz-berlin.de
*   Correspondence: martin.beye@desy.de

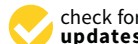

**Featured Application: Exploiting the full flux and temporal resolution of SASE-FELs for highly sensitive X-ray absorption measurements.**

**Abstract:** X-ray absorption spectroscopy (XAS) and the directly linked X-ray reflectivity near absorption edges yield a wealth of specific information on the electronic structure around the resonantly addressed element. Observing the dynamic response of complex materials to optical excitations in pump–probe experiments requires high sensitivity to small changes in the spectra which in turn necessitates the brilliance of free electron laser (FEL) pulses. However, due to the fluctuating spectral content of pulses generated by self-amplified spontaneous emission (SASE), FEL experiments often struggle to reach the full sensitivity and time-resolution that FELs can in principle enable. Here, we implement a setup which solves two common challenges in this type of spectroscopy using FELs: First, we achieve a high spectral resolution by using a spectrometer downstream of the sample instead of a monochromator upstream of the sample. Thus, the full FEL bandwidth contributes to the measurement at the same time, and the FEL pulse duration is not elongated by a monochromator. Second, the FEL beam is divided into identical copies by a transmission grating beam splitter so that two spectra from separate spots on the sample (or from the sample and known reference) can be recorded in-parallel with the same spectrometer, enabling a spectrally resolved intensity normalization of pulse fluctuations in pump–probe scenarios. We analyze the capabilities of this setup around the oxygen *K*- and nickel *L*-edges recorded with third harmonic radiation of the free electron laser in Hamburg (FLASH), demonstrating the capability for pump–probe measurements with sensitivity to reflectivity changes on the per mill level.

**Keywords:** X-ray absorption spectroscopy; free electron laser; intensity normalization; transmission grating; X-ray reflectivity

## 1. Introduction

X-ray absorption spectroscopy (XAS) is a common method for characterizing materials in a variety of fields due to its fundamental simplicity, flexibility and element-specificity. Tuning to absorption edges allows probing the unoccupied electronic states localized around specific elements. Although experimentally more challenging, this information content is especially high at the absorption edges in the soft X-ray regime [1], since core hole lifetimes are longer and spectral features sharper than at higher photon energies. In a pump–probe experiment with sufficiently short pulses, e.g., in free electron lasers (FELs), transient XAS can be used to track changes of the electronic structure during chemical reactions and phase transitions on the femtosecond (fs) to picosecond (ps) scale [2–4].

XAS measures the spectral absorption coefficient, i.e., the imaginary part of the index of refraction. Measurements in specular reflectivity additionally measure the real part of the refractive index, which is rigorously connected with the imaginary part, through the Kramers–Kronig transform [5]. Thus, the same spectroscopic information is transported [6]. The most straightforward method of measuring XAS is in transmission by monitoring the intensity of an incident and the transmitted monochromatic beam as a function of photon energy. However, short absorption lengths in the soft X-ray regime necessitate sub-μm thin samples, which can be a prohibitive limitation in terms of properties and manufacturability of the sample. Therefore, XAS is often performed indirectly, by observing either electron yield or drain current resulting from the photoelectric effect or the fluorescence yield. Each of these methods has advantages and disadvantages. Broadly speaking, electron-based methods constitute a powerful approach but limit the obtained information to the surface region and suffer from space-charge and capacitance effects when implemented with intense pulsed sources. Fluorescence based methods are free of such charge effects but struggle from low signal levels in the soft X-ray regime, as the Auger process is dominant and suppresses radiative decay. Furthermore, additional selection rules constraining fluorescent decay as well as competing fluorescence channels can lead to deviations in the spectra that can only be fully interpreted and disentangled [7–9] after spectral analysis of the isotropically emitted fluorescence from the sample, using a spectrometer. As in the soft X-ray regime, spectrometers operate with gratings at grazing incidence angles, such spectrometers exhibit small solid acceptance angles, strongly reducing the overall detected signal. Therefore, although a multitude of these methods are applied with great success at synchrotron sources, performing XAS studies at FEL facilities [2,10,11] remains challenging. Another challenge particular to FELs, is the stochastic nature of radiation from the self-amplified spontaneous emission (SASE) process on which most FELs rely (except those that implement seeding). The X-ray pulses produced by the SASE process exhibit a number of Fourier-limited modes of random intensity, which are randomly distributed within overall pulse durations typically on the order of 50 fs and a spectral bandwidth of typically 0.5–1% [12]. Using a monochromator to gain higher spectral resolution enhances the strong intensity fluctuations of the resulting beam and discards a significant part of the incoming flux. For XAS measurements, this means that the incident flux cannot be approximated as constant, but must be measured with the same fidelity, sensitivity and dynamic range as the signal. Furthermore, monochromators can significantly elongate the FEL pulse duration due to grating induced pulse-front tilting, which scales with the number of illuminated grating lines and is thus especially severe (up to a picosecond) at very high spectral resolutions.

Here, we demonstrate the use of a transmission grating beam-splitter to split the FEL into practically identical signal and reference beams, both of which are analyzed in-parallel with the same spectrometer after interaction with the sample. Thus, both FEL fluctuations, as well as possible nonlinearities in the detector are exactly reproduced in both signals and can thus be renormalized.

Unlike comparable schemes with monochromatic beams [13–16], our method places the grating for spectral analysis after the sample interaction, so that the full SASE bandwidth contributes to the measurement. For a given FEL fluence on the sample, measuring the collimated specular reflection as opposed to isotropic fluorescence provides much stronger signals. The signal intensity can even be tuned by adjusting the angle of incidence to optimally exploit the detector dynamic range, since varying the angle will generally alter the spectral shape but not change the spectroscopic information. Here, we analyze the sensitivity of this experimental scheme for spectral (e.g., pump-induced) changes by evaluating pairs of simultaneously recorded reflectivity spectra of NiO at the oxygen $K$- and nickel $L_{2,3}$-edges, respectively.

## 2. Experimental Design

A simplified schematic of the setup is shown in Figure 1. Measurements were performed with the MUSIX experimental end station [17] at the FL24 beam line of the free-electron laser in Hamburg (FLASH). The FEL was tuned to generate third harmonic radiation around the oxygen $K$- and nickel $L_{2,3}$-edges (from 506 eV to 566 eV and 842 eV to 887 eV, respectively), producing bursts of 40 pulses with 10 µs spacing within the burst and 10 bursts per second. Before the beam line, the beam is defined by two apertures: 5 mm apertures are used at the O $K$-edge and 2 mm apertures at the Ni $L$-edges. Then, the average FEL pulse energy was monitored using the signal from an X-ray gas monitor detector (XGMD) [18], after which the fundamental radiation was suppressed by a 13 meter long gas attenuator unit containing $9.7 \times 10^{-2}$ mbar of neon (O $K$-edge) or $1.7 \times 10^{-2}$ mbar of krypton (Ni $L$-edges) in addition to two Si membrane filters of 401 nm and 200 nm thickness.

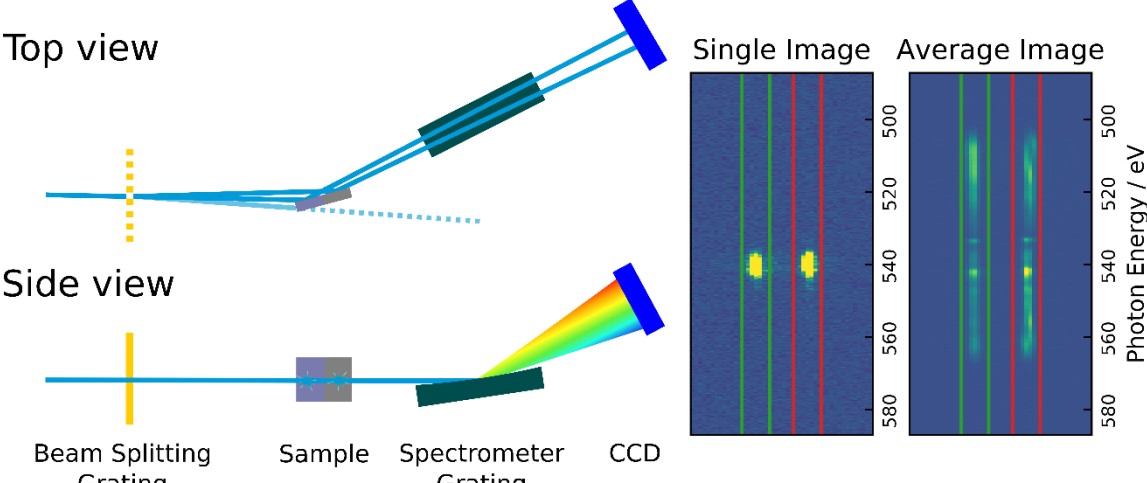

**Figure 1.** Schematic setup Experimental setup at the FL24 beam line of the free electron laser in Hamburg (FLASH) (left) and exemplary detector images (right). The free electron laser (FEL) beam is initially transmitted through a beam splitting grating, optimized for high intensity in the first diffraction orders. The zeroth and one of the first diffraction orders are then focused onto the sample with a vertical focal size of about 35 µm at a grazing incidence of 11.5 °. The specular reflection of both beams is directed onto the variable line spacing diffraction grating of the spectrometer and dispersed onto a charge-coupled device (CCD). Since the spectrometer disperses both beams orthogonally to the beam separation, the two beams yield separate spectra on the detector. The green and red lines mark the regions of interest in which spectral intensity is integrated and correspond to the spectra in Figure 2a. 1208 single images are averaged while the undulators are scanned across the O $K$-edge with both beams on the NiO sample. The average image shows the spectral structure within the scanned spectral interval.

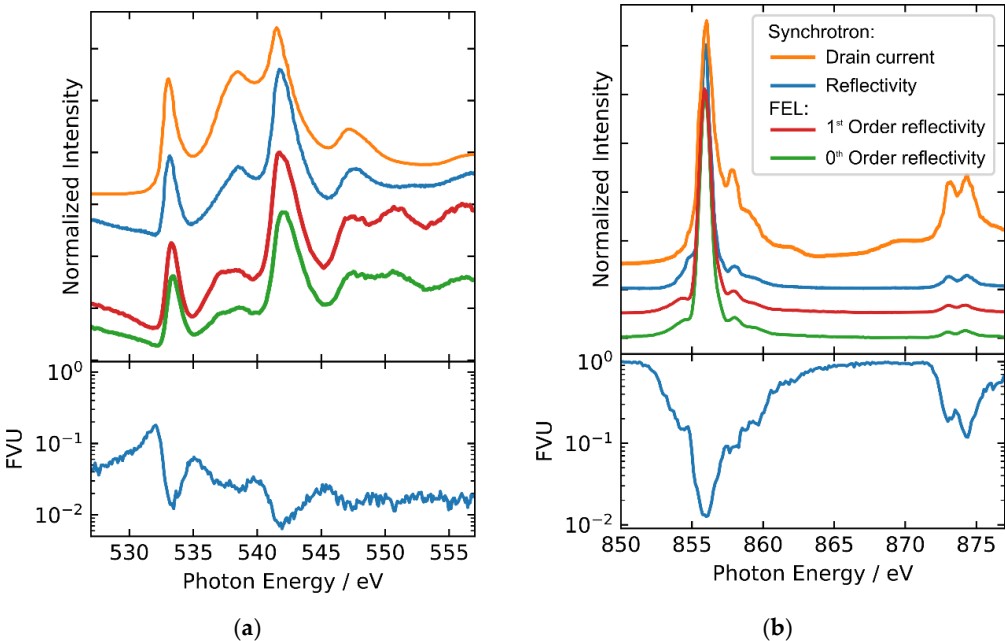

**Figure 2.** Reflectivity spectra pairs (red and green) recorded in-parallel at FLASH including the spectrally resolved fraction of variance unexplained (FVU) between the FEL spectra. Reflectivity (blue) and drain current (orange) spectra recorded at a synchrotron are given for comparison. The FEL spectra, each pair acquired within 4 minutes, are normalized so that their relative intensity is conserved. A constant shift in photon energy is applied to match the calibration of the synchrotron spectra. Reference spectra are recorded from the same samples at the beam line PM3 of BESSY II. (**a**) Oxygen *K*-edge. Additional structure from water residue on the surface contributes additional intensity, mostly to the high energy side (>545 eV) of the spectra since the samples were exposed to air before both FEL and synchrotron measurements. (**b**) Ni $L_3$-and $L_2$-edges.

The beam is then split horizontally by a transmission grating of 3.8 mm height and 0.9 mm width, optimized to produce similar intensity in the zeroth and first diffraction orders: It is made from a Si-membrane of 1.1 μm thickness with rectangular grooves of 960 nm depth and 7.9 μm period at 50% aspect ratio. The zeroth and first diffraction orders are each directed onto the sample, focused by bendable mirrors to a spot of about 35 μm height and 80 μm width, at a grazing angle $\theta$ of 11.5°. The specular reflections of both beams are analyzed by the spectrometer of the MUSIX endstation [17], consisting of a variable line spacing grating and an in-vacuum charge-coupled device (CCD) (GreatEyes Model GE-VAC 2048 2048). The 13.5 μm square pixels of the CCD were read out with an ADC clocked with 3 MHz in high gain mode and 16 pixel binning orthogonal to the spectral axis to increase the frame rate and decrease noise while preserving the spectral resolution. This resulted in a framerate of 5 Hz. The FEL, producing bursts of 10 Hz, was chopped accordingly, so that each detector image represented an average over one burst of 40 pulses. The photon energy dispersion on the CCD was measured as 10.73 pixel/eV at the O *K*-edge and 7.8 pixel/eV at the Ni *L*-edges.

Samples consisted of epitaxial NiO films of 40 nm thickness, grown on MgO(001) substrates. A 2 nm thick MgO underlayer was deposited by radio frequency magnetron sputtering in 3 mTorr argon at a temperature below 100 °C followed by a NiO layer that was deposited at 700 °C in an Ar (90%)/$O_2$ (10%) gas mixture at a pressure of 3 mTorr and was then annealed in-situ at the same temperature (700 °C) for 15 min in the same Ar-$O_2$ gas mixture.

To acquire spectra within a wider window than the 0.5–1% natural bandwidth of the FEL, the photon energy is scanned in steps of 0.75 eV by varying the undulator gap. As datasets include multiple scans over the desired range, data taken at the same undulator settings are averaged in the data analysis. Two spectra are extracted from each image by integration along the non-dispersive direction of the CCD within manually selected regions of interest such as indicated in Figure 1. Before

computing the average spectrum, each single-image spectrum was cropped to the range deviating less than 1 % from the FEL photon energy set-point, as no significant intensity is found outside this window.

## 3. Results and Discussion

To understand the benefits of acquiring two spectra in-parallel, we consider the measured spectral intensity on the detector $S(\omega)$, which depends on the photon frequency $\omega$ for each beam, to be proportional to the spectral reflectivity $R(\omega)$ of the sample, the diffraction efficiency $O$ of the beam-splitting grating in the zeroth or first order and the spectral intensity $I(\omega)$ of the FEL:

$$S_1(\omega) \propto R_1(\omega)O_1 I(\omega) \quad S_2(\omega) \propto R_2(\omega)O_0 I(\omega) \tag{1}$$

It is apparent that the measured spectrum $S(\omega)$ of a single beam can be used to measure the reflected spectrum $R(\omega)$, if the FEL intensity $I(\omega)$ is either known or constant. If many pulses are averaged, SASE fluctuations average out, but potential slow drifts of the average pulse energy remain. Since the third harmonic radiation intensity scales roughly with the square of the fundamental [17], slow fluctuations are mitigated by normalizing each spectrum with the square intensity measured by the XGMD which monitors the fundamental radiation of the FEL further upstream. This leads to the FEL spectra shown in Figure 2a,b, each pair was measured in-parallel using $10^5$ SASE pulses within four minutes. Synchrotron spectra of reflectivity and drain current recorded on the same sample with a similar angle of incidence ($\theta = 11°$ for the O $K$-edge, $\theta = 12°$ for the Ni $L_{2,3}$-edges) are shown for comparison and demonstrate a good agreement to the FEL spectra. The FEL-reflectivity spectra recorded at these angles reproduce the relevant structures of the drain current absorption spectra. We ascribe the minor differences to sample inhomogeneity and the difference of the incident angle. Especially the O $K$-edge spectra also show signs of surface contamination with water (especially the structure above 545 eV), as the sample was exposed to air both before and between FEL and synchrotron measurements. In the following, we analyze how accurate changes in one spectrum due to FEL fluctuations can be used to measure the changes in the other spectrum. Our main motivation for recording two spectra in-parallel is to exploit one of them as a spectrally resolved intensity reference in a pump–probe scenario. In this case, a crucial parameter is the precision with which (pump-induced) changes in one spectrum may be determined from the other. The fraction of fluctuation that cannot be explained from the variations in the reference spectrum is quantified by the so-called fraction of variance unexplained (FVU). The FVU for a set of $N$ pairs of spectra equal unity minus the coefficient of determination $r^2$ between both spectra, which is in this case, the square of the Pearson correlation coefficient $r$:

$$r(\omega) = \frac{\sum_{i=1}^{N}\left(S_1^i(\omega) - \overline{S}_1(\omega)\right)\left(S_2^i(\omega) - \overline{S}_2(\omega)\right)}{\sqrt{\sum_{i=1}^{N}\left(S_1^i(\omega) - \overline{S}_1(\omega)\right)^2}\sqrt{\sum_{i=1}^{N}\left(S_2^i(\omega) - \overline{S}_2(\omega)\right)^2}} \tag{2}$$

$$FVU = 1 - r^2 \tag{3}$$

Here, $\overline{S}$ denotes the average spectrum over the set. We find that minor alignment imperfections and pointing drifts can lead to a small offset along the dispersive direction of the detector between two simultaneously recorded spectra. Similar to previous work [15], this offset is determined by shifting the reference spectrum along the dispersive axis (see Appendix A for details) such that the logarithmic FVU, integrated over the entire spectrum, is minimized. The offset is determined in this way for every measurement (each in the order of several minutes) separately, yielding a shift between one and three pixels, i.e., some tens of μm on the CCD detector. The optimized spectrally resolved FVU is shown in Figure 2 and reproduces the structure of the spectrum as it scales inversely with the reflected intensity. The scaling of the FVU with intensity is shown in Figure 3a and is compared to a simulation of the FVU which may be expected from of a Poisson-distributed noise process, scaling with the square root of the intensity like a photon shot noise (orange), and an additional normally-distributed noise process with constant variance like readout noise (green). Since the fast readout of the CCD prevented

a calibration of the sensitivity from single-photon incidences, the number of photons per detector count was estimated based on the assumption that the noise level at high intensities is dominated by photon shot noise, as was found in a similar setup before [15]. This assumption is supported by the scaling behavior shown in Figure 3a and is in rough agreement with the manufacturer specifications of the CCD.

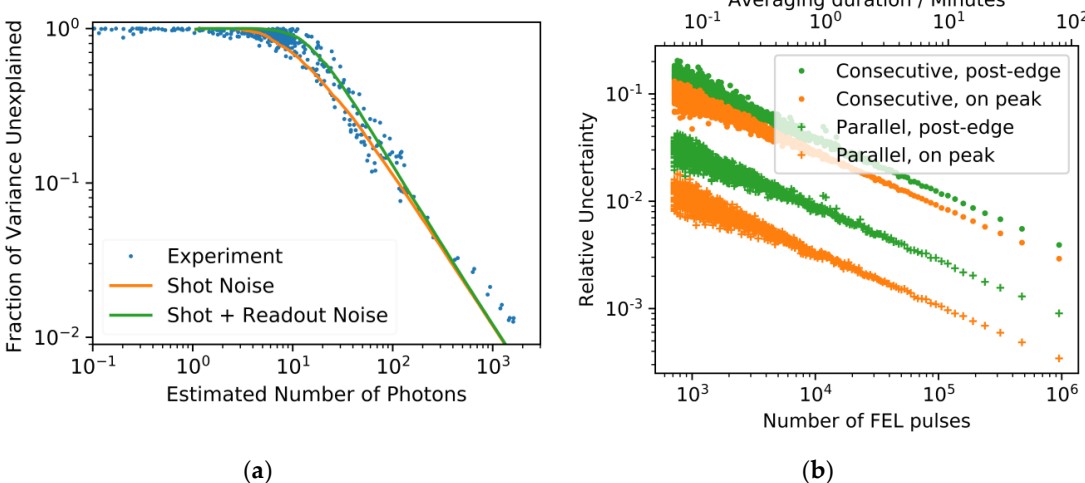

(**a**)　　　　　　　　　　　　　　　　　　　　(**b**)

**Figure 3.** Evaluations of the setup sensitivity: (**a**) The fraction of variance unexplained, (FVU, see Equation 3) between the two FEL-spectra shown in Figure 2b plotted against intensity, given as an estimated number of detected photons within a given spectral interval. We show experimental data points (blue) along with a simulation that accounts for photon shot noise only (orange), as well as additionally including a Gaussian readout noise contribution (green). (**b**) Uncertainty to spectral changes, such as pump–probe effects, considering two methods of acquisition. The plot shows the uncertainty of the average ratio between intensities at the Ni $L_3$-edge (orange) and 1 eV beyond the edge (green, each within a 245 meV window), measured with the FEL photon energy at the $L_3$-edge. The uncertainty for parallel acquisition (crosses) considers the correlation between the two intensities and thus, yields an order of magnitude lower uncertainties for this measurement mode. The consecutive acquisition uncertainty (circles) is calculated from the same data, assuming no correlation between the intensity fluctuations.

Figure 3b further illustrates the uncertainty with which the ratio between the reflectivity probed by both beams may be determined with increasing acquisition time. This uncertainty is the precision with which pump–probe changes in one spectrum could be detected. To this purpose, the undulators are tuned to the Ni $L_3$-edge. Without scanning the undulators, the ratio between the intensities in both spectra was evaluated. Two exemplary regions in the spectra were analyzed: First, a 245 meV region around the $L_3$-edge peak at 857 eV (corresponding to two rows on the detector). Second, an equally sized window from the same dataset was evaluated 1 eV above the peak, where the detected intensity was lower by about a factor of nine on average, due to both the lower reflectivity and the FEL being tuned to the $L_3$-edge. The plot shows the relative uncertainty with which the ratio may be determined for increasingly larger subsets of CCD images, randomly drawn (without replacing) from an 80-min measurement. Two different measures for the uncertainty are shown: First, the uncertainty regarding the parallel acquisition, which considers that the correlation between the two intensities reduces the uncertainty for the estimator of the mean. This error propagation was discussed previously [15] and is reiterated in Appendix B. Second, the error is calculated under the assumption that the fluctuations of both intensities are uncorrelated, which would be the case if the spectra had been recorded consecutively instead of in-parallel.

## 4. Conclusions

As demonstrated in Figure 3b, the acquisition of two spectra in-parallel merits about one order of magnitude of sensitivity compared to consecutive acquisition. Spectral changes can be monitored within the entire FEL bandwidth at the same time. The sensitivity in the middle of the FEL spectrum reaches a $10^{-3}$ relative reflectivity change within about 10 min of acquisition for a 245 meV window. The reflectivity spectra acquired in this way reproduce the features of conventional X-ray absorption spectra, so that equivalent information may be gained by pump–probe experiments in reflectivity. As the spectrometer grating analyzes the full reflected beams after sample interaction, the temporal resolution is not diminished by monochromatization. This makes the presented setup ideally suited for time resolved XAS and reflectivity studies in FELs.

**Author Contributions:** Conceptualization, P.S.M., C.S.-L., H.A.D. and M.B.; Data curation, R.Y.E., D.T. and I.V.; Formal analysis, R.Y.E.; Funding acquisition, H.A.D. and M.B.; Investigation, R.Y.E., P.S.M., D.T., I.V., G.B., S.D., M.K., H.A.D. and M.B.; Methodology, R.Y.E., F.D., C.D., C.S., H.A.D. and M.B.; Project administration, M.B.; Resources, F.D., Stuart S.P. Parkin, C.D., C.S.-L., H.A.D. and M.B.; Software, R.Y.E. and A.S.; Supervision, M.B.; Validation, M.B.; Visualization, R.Y.E.; Writing—original draft, R.Y.E.; Writing—review and editing, R.Y.E., J.O.S., C.S.-L., H.A.D. and M.B. All authors have read and agreed to the published version of the manuscript.

**Funding:** M.B., P.S.M., J.O.S. and R.Y.E. received funding from the Helmholtz Association through grant VH-NG-1105. IV acknowledges support by the U.S. Department of Energy, Office of Science, Office of Basic Energy Sciences under the X-Ray Scattering Program Award Number DE-SC0017643. Work at UU was supported by the Swedish Research Council (VR) grants 2017-06711 and 2018-04918. F.D. received funding from the EU-H2020 Research and Innovation Programme under the Marie Skłodowska-Curie grant agreement No. 701647.

**Acknowledgments:** We acknowledge DESY (Hamburg, Germany) and the Helmholtz-Zentrum Berlin, both members of the Helmholtz Association HGF, for the provision of experimental facilities. FEL measurements were carried out at the FL24 beam line of FLASH. Synchrotron measurements were carried out at the PM3 beam line of BESSY II, operated by HZB. The samples were prepared at the MPI-Halle by A.S. and S.S.P.P.

**Conflicts of Interest:** The authors declare no conflict of interest. The funders had no role in the design of the study; in the collection, analyses, or interpretation of data; in the writing of the manuscript, or in the decision to publish the results.

## Appendix A

Shifting spectra with sub-pixel accuracy was achieved by resampling the spectrum with a 100-fold frequency using the resample function of the python library scipy [19]. A Gaussian window function with a width of half the number of points in the original spectrum was applied to suppress high-frequency components arising in the oversampling process. The oversampled spectra were then shifted by an integer number of points, such that the logarithmic FVU is minimized. Finally, the spectrum is resampled back to the original spectral axis using the same resample function.

## Appendix B

The uncertainty of the expectation value of the ratio between two spectral intensities which were measured in-parallel, i.e., not independent of each other, can be propagated as follows.

$$\sigma_T = \left| \frac{\langle I_2 \rangle}{\langle I_1 \rangle} \right| \sqrt{ \left( \frac{\sigma_2}{\langle I_2 \rangle} \right)^2 + \left( \frac{\sigma_1}{\langle I_1 \rangle} \right)^2 - \frac{2\,\sigma_{I_2 I_1}}{\langle I_2 \rangle \langle I_1 \rangle} }$$

Here, $\sigma_1$ and $\sigma_2$ represent the uncertainty of the expectation value (i.e., the standard deviation divided by the square root of the number of measurements) for the intensities $I_1$ and $I_2$ and $\sigma_{I_2 I_1}$ is the covariance between both. Brackets $\langle\,\rangle$ denote the ensemble mean value.

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
