# Peer review of "Parallel Broadband Femtosecond Reflection Spectroscopy at a Soft X-Ray Free-Electron Laser"

_applsci, doi:10.3390/app10196947_

Round 1

Reviewer 1 Report

This work demonstrates that the acquisition of two spectra in parallel improves the sensitivity of X-ray absorption spectroscopy (XAS) by an order of magnitude, with regards to consecutive acquisition. This is an outstanding result for time resolved XAS and reflectivity studies in free electron lasers. I consider that this paper is an excellent match for Applied Sciences, since this study is very relevant for the research community, as well as for the field of materials characterization. Additionally, it is well written, includes high-quality, clear and relevant figures and the explanations in the text yield to the results content in a proper way. The mathematical foundations included are correct to the best of my knowledge and the reference section is quite complete as well. All in all, I consider this work very well done and I recommend its publication unaltered.

Author Response

Dear reviewer,

thank you very much for your positive review!

Many Regards,

Robin Engel

Reviewer 2 Report

REFEREE REPORT

on the manuscript by R. Y. Engel et al. “Parallel broadband femtosecond reflection spectroscopy at a soft X-ray free-electron laser”.

The manuscript is devoted to a realization at the FL24 beam line of the free electron laser (EEL) in Hamburg (FLASH) a setup for reflection spectroscopy. Instead of the use of a monochromator in a beam line before a studied sample they employ a transmission grating beam-splitter to split the EEL into three identical beams in zero and first diffraction orders. The zero and one of the first diffraction orders are directed to a sample and then the reflected spectra are analysed in parallel by a grazing incidence spectrometer. Such scheme has two main advantages: one beam gives a signal, while the other permits a spectrally resolved intensity normalization of pulse fluctuations, and simultaneously the full self-amplified spontaneous emission bandwidth is measured. The potential of the setup is demonstrated by recording and analysing of the reflectivity spectra around the oxygen K- and nickel L-edges. It was shown that the recording of two spectra in parallel has about one order of magnitude higher sensitivity than in consecutive recording scheme.

The results are new and clearly presented. The manuscript can be accepted without any changes.

Author Response

Dear reviewer,

thank you very much for the accurate summary and the positive review!

Many regards,

Robin Engel

Reviewer 3 Report

The article presents the results of the very interesting and important for future pump-probe investigations experiments. X-ray reflectivity near absorption edges has enormous perspectives of applications in different scientific regions. The authors suggest the way how to overcome the fluctuating spectral content of pulses generated by self-amplified spontaneous emission (SASE), which in some way makes impossible the dynamical spectral analysis.

The article definitely should have great interest for readers and should be published.

Anyhow I would add some comments.

  1. the FEL beam is divided into identical copies by a transmission grating beam splitter”. They are obtained by “the zeroth and first diffraction orders” of reflection. I have not fount the experimental proof of their identity. To my opinion the diffraction changes the spectral content of the pulses and (which is more important) in different way for the different orders of diffraction. In other words, the O1 and O0 parameters in (1) should include different energy (ω) dependences. Probably the difference of the measured spectrum pairs can be explained by this way.
  2. I am slightly confused by neglecting difference of the absorption and reflectivity spectra in the article. “The reflectivity spectra acquired by this way reproduce the structure of conventional X-ray absorption spectra.” And the following suggestion: “The signal intensity can even be tuned by varying the angle of incidence to optimally exploit the detector dynamic.” The angle variations lead not just to the intensity variation! I remember the very old illustration of the difference of the absorption and reflectivity spectra from -C. Kao et al., PRB 50, 9599 (1994) (picture included in pdf report)
  1. Figure 1: Do red and green vertical lines in the images show the limits for averaging? It is not mentioned.
  2. I have not found the NiO film thickness which is important for the reflectivity behavior.
  3. The sentence

the additional transition operators involved in the decay (e.g. adding dipole selection rules) lead to deviations in the spectra that can only be fully interpreted and disentangled [7–9] after spectral analysis of the isotropically emitted fluorescence from the sample, using a spectrometer. “

is not quite clear. The authors mean the multipolarity or anisotropy of transitions? But I guess not “operators”.

In spite of my comments I think that the article should be published

Author Response

Dear reviewer,

thank you very much for your review and for your questions.
Please find our answers below and the modifications to the manuscript in the updated word file with active “track changes”.

Figures are included in the word document version of this reply.

Comment: “the FEL beam is divided into identical copies by a transmission grating beam splitter”. They are obtained by “the zeroth and first diffraction orders” of reflection. I have not found the experimental proof of their identity. To my opinion the diffraction changes the spectral content of the pulses and (which is more important) in different way for the different orders of diffraction. In other words, the O1 and O0 parameters in (1) should include different energy (ω) dependences. Probably the difference of the measured spectrum pairs can be explained by this way.

The reviewer raises the question if the spectral content of the diffraction orders is truly identical, since diffraction is generally a wavelength-dependent phenomenon.
Due to the reflection geometry chosen in this experiment, it was not possible to record spectra without any sample. Nevertheless, a measurement of single FEL pulses without sample was done in transmission by part of our collaboration (Brenner et. al., Opt. lett. 44.9 (2019): 2157-2160.10.1364/ol.44.002157, Figure 2), using a similar beam splitting grating. A constant difference in diffraction efficiency is visible between first and zeroth diffraction order, but spectral structures are accurately reproduced in all orders, limited by shot noise and readout noise.
In the presented measurements, the grating was designed such that the intensity ratio of first and zeroth diffraction orders was close to unity.

The wavelength dependency of the diffraction angle that the reviewer points out leads to the first orders of diffraction being angled slightly with respect to the zeroth order, as longer wavelengths experience a larger diffraction angle. However, due to the very low line density of the beam splitting grating, this effect only becomes noticeable at all when the photon energy range on the detector becomes a significant fraction of the overall photon energy, which is rarely the case in pump probe measurements. It is slightly noticeable in the first order spectrum shown in figure 1.
Even then, the wavelength-dependency of the splitting is of no consequence for the final spectra, as this splitting occurs orthogonally to the dispersive direction of the spectrometer and the signal is integrated within the ROIs (marked in the Figure) along this direction.

For these reasons, we decided to treat O0 and O1 as constants for the purposes of this work.

Comment: I am slightly confused by neglecting difference of the absorption and reflectivity spectra in the article. “The reflectivity spectra acquired by this way reproduce the structure of conventional X-ray absorption spectra.” And the following suggestion: “The signal intensity can even be tuned by varying the angle of incidence to optimally exploit the detector dynamic.” The angle variations lead not just to the intensity variation! I remember the very old illustration of the difference of the absorption and reflectivity spectra from -C. Kao et al., PRB 50, 9599 (1994) (picture included in pdf report)

The points raised by the reviewer here are fully correct. By no means are absorption and reflectivity spectra identical. We included an electron yield spectrum in the synchrotron spectra shown in Figure 2, which is an absorption spectrum, showing that the features present in the absorption spectrum are also present in the reflectivity measurements, albeit at different intensities.
We avoided a lengthy discussion of the difference in this work, since we intended to focus on the aspects connected to the parallel acquisition scheme and the sensitivity reachable in such a scheme. For this purpose, pump-probe measurements in reflectivity and absorption probe the same features of the electronic structure. The Fresnel- and Kramers-Kronig relations also provide a rigorous relationship between the results of both measurements.
To avoid arousing the same confusion in our readers, we adapt the quoted passages:

“The reflectivity spectra acquired by this way reproduce the structure of conventional X-ray absorption spectra.”

“The reflectivity spectra acquired in this way reproduce the features of conventional X-ray absorption spectra, so that equivalent information may be gained by pump-probe experiments in reflectivity.

“The signal intensity can even be tuned by varying the angle of incidence to optimally exploit the detector dynamic.”

“The signal intensity can even be tuned by adjusting the angle of incidence to optimally exploit the detector dynamic range, since varying the angle will generally alter the spectral shape but not change the spectroscopic information.”

Comment: Figure 1: Do red and green vertical lines in the images show the limits for averaging? It is not mentioned.

Yes. We now mention it.

Comment: I have not found the NiO film thicknesswhich is important for the reflectivity behavior.

Thank you for pointing out this oversight.

“Samples consisted of epitaxial NiO films grown on MgO(001) substrates.”

“Samples consisted of epitaxial NiO films of 40 nm thickness, grown on MgO(001) substrates.”

Comment: The sentence
“the additional transition operators involved in the decay (e.g. adding dipole selection rules) lead to deviations in the spectra that can only be fully interpreted and disentangled [7–9] after spectral analysis of the isotropically emitted fluorescence from the sample, using a spectrometer. “
is not quite clear. The authors mean the multipolarity or anisotropy of transitions? But I guess not “operators”.

Thank you for pointing out this imprecise wording.

“the additional selection rules constraining fluorescent decay as well as competing fluorescence channels can lead to deviations in the spectra that can only be fully interpreted and disentangled [7–9] after spectral analysis of the fluorescence from the sample, using a spectrometer. “
